# Nanoscale Zero-Valent Iron and Chitosan Functionalized *Eichhornia crassipes* Biochar for Efficient Hexavalent Chromium Removal

**DOI:** 10.3390/ijerph16173046

**Published:** 2019-08-22

**Authors:** Xue-Li Chen, Feng Li, Xiao Jie Xie, Zhi Li, Long Chen

**Affiliations:** 1School of Civil Engineering & Transportation, South China University of Technology, Guangzhou 510640, China; 2San Bernardino, California State University, San Bernardino, CA 92407, USA; 3Department of Civil and Environmental Engineering, Northeastern University, Boston, MA 02115, USA

**Keywords:** chromium, biochar, nanoscale zero-valent iron, chitosan, sorption

## Abstract

Sorption is widely used for the removal of toxic heavy metals such as hexavalent chromium (Cr(VI)) from aqueous solutions. Green sorbents prepared from biomass are attractive, because they leverage the value of waste biomass and reduce the overall cost of water treatment. In this study, we fabricated biochar (BC) adsorbent from the biomass of water hyacinth (*Eichhornia crassipes*), an invasive species in many river channels. Pristine BC was further modified with nanoscale zero-valent iron (nZVI) and stabilized with chitosan (C) to form C–nZVI–BC. C–nZVI–BC adsorbent showed high hexavalent chromium sorption capacity (82.2 mg/g) at pH 2 and removed 97.34% of 50 mg/L Cr(VI) from aqueous solutions. The sorption capacity of chitosan–nZVI-modified biochar decreased while increasing the solution pH value and ionic strength. The results of a sorption test indicated that multiple mechanisms accounted for Cr(VI) removal by C–nZVI–BC, including complexation, precipitation, electrostatic interactions, and reduction. Our study suggests a way of adding value to biomass waste by considering environmental treatment purposes.

## 1. Introduction

Chromium (Cr) is widely used in corrosion-resistant material and inorganic pigments [1,2], and the improper release of Cr has caused severe environmental pollution [3,4,5]. The Cr ion has two valences, i.e., the trivalent Cr(III) and hexavalent Cr(VI) valences. Compared to Cr(III), Cr(VI) shows higher acute toxicity and has been categorized as a carcinogenic substance by the U.S. Agency for Toxic Substances and Diseases Registry (ATSDR) [6]. Moreover, Cr(VI) is stable in complicated water chemistry and is soluble over a wide pH range [2,4,7], leading to severe health concerns.

Sorption has been demonstrated as an economic and effective way to remove many toxic compounds and ions, mainly because the overall process is easy to operate and no secondary contaminants are generated [8,9,10,11,12,13]. For the past few decades, various sorbents have been developed, including activated carbon [8], clay minerals [9], metal–organic frameworks [11], and two-dimensional nanomaterials [10,13]. Recently, an amorphous and porous carbon material produced from the pyrolysis of biomass, named biochar, has shown strong potential in environmental remediation [12,14,15]. For example, biochars prepared from soybean stalks [16], bamboo and rice straw [17], and sugarcane bagasse [18] were used to remove various heavy metals from aqueous solution, with remarkable efficiencies comparable to activated carbon [8]. The most significant advantages of biochar as an adsorbent are low costs, its abundancy as a raw material, and easiness to produce [12,14].

Water hyacinth (*Eichhornia crassipes*) is an exotic plant that has invaded many rivers around the world, and the uncontrolled growth causes safety issues to the sailing industry and ecological concerns to local fisheries [19,20,21]. The harvested *E. crassipes* biomass requires complicated deposition steps for treatment [20]. The conversion of *E. crassipes* biomass into an adsorbent is a sustainable strategy to simultaneously address biomass deposition and environmental pollution [22,23,24,25,26,27]. Due to its rapid growth, *E. crassipes* biomass is relatively simple in composition, which enables its pyrolysis into biochar [23,24,25,26]. However, pristine biochar usually owns unsatisfactory sorption capability, and further modifications are needed [14,28]. The loading of nanoscale zero-valent irons (nZVIs) on the biochar surface is an effective way to enhance heavy metal removal, based on reported synergy mechanisms including reduction, electrostatic attraction, and sorption [3,29,30]. In addition, cation exchange resin-supported nano-iron materials also have been used for the treatment of Cr(VI)-contaminated groundwater [31]. However, the major challenges of nZVI modification include agglomeration due to magnetic attraction and oxidation after exposure to air [3]. It has been reported that nZVI could be stabilized by assembling multiple-layer materials. For example, chitosan is able to complex with nZVI and reduce interparticular attractions.

In this work, a multilayered chitosan–nZVI-modified biochar (C–nZVI–BC) was prepared via sequentially growing nZVI particles and adding chitosan-stabilizing agent to pristine BC. The C–nZVI–BC composite showed high sorption efficiency toward Cr(VI). The structure, morphology, compositional distribution, and element chemical states of C–nZVI–BC were extensively characterized using X-ray diffraction (XRD), scanning electron microscopy (SEM), energy-dispersive X-ray spectroscopy (EDS), and X-ray photoelectron spectroscopy (XPS). A synergy of multiple mechanisms is proposed to account for the excellent Cr(VI) removal efficiency in our study. Our study suggests modification strategies for future biochar design based on revealed mechanisms.

## 2. Materials and Methods 

### 2.1. Materials

Fresh *E. crassipes* biomasses were collected from a stream in Nansha district, Guangzhou, China. Hexavalent chromium standard solution was purchased from Beijing Aike Yingchuang Chemical Reagent Co., Ltd., China (standard substances network). Other chemicals, including FeSO_4_·7H_2_O (99%, analytical reagent grade), polyethylene glycol (PEG-4000, guaranteed reagent grade), NaBH_4_ (98%, analytical reagent grade), chitosan (90%, guaranteed reagent grade), glutaraldehyde (25%, guaranteed reagent grade), NaOH, acetic acid, anhydrous ethanol, and hydrochloric acid were of analytical reagent grade from Tansoole Chemical Technology (Shanghai, China) and were used as received without further purification.

### 2.2. Preparation of Adsorbents

#### 2.2.1. Preparation of Pristine Biochar (BC)

The *E. crassipes* biomasses were cleaned, cut into 5-cm pieces, and then dried at 95 °C. The dried biomasses (300 g) were heated to 400 °C in a pipe furnace with nitrogen as the fluidizing gas at a rate of 1.5 L/min and then held for 3 h. In order to remove the ash of biochar, the biochar was soaked in 25% HCl for 12 h with stirring. The biochar was subsequently washed with alcohol and deionized water each for three times and then dried in an oven at 95 °C. The obtained biochar was grounded and sieved to 0.5~1-mm-sized particles for future use. The resulting biochar was referred to as BC.

#### 2.2.2. Preparation of nZVI–BC Composite

Under a nitrogen atmosphere, FeSO_4_·7H_2_O (1.0 mmol, 287 mg), polyethylene glycol (PEG-4000, 0.5 g) dispersant, and BC (6 g) were added to 70% aqueous ethanol (200 mL) and stirred at room temperature for 3 h. Then, NaBH_4_ solution (50 mL at 0.4 mol/L) was added dropwise into the above slurry and was kept stirred at room temperature for 1 h. Black powder was subsequently collected by filtration and then washed three times with deionized water and anhydrous ethanol. The pellets were then vacuum-dried at −25 °C for 24 h. The obtained product was sieved through 0.5~1-mm screens and stored for further use. The resulting material was labeled as nZVI–BC.

#### 2.2.3. Preparation of C–nZVI–BC Composite

Here, 3 g chitosan was dissolved in 200 mL diluted acetic acid solution (2%, v/v). After the addition of 3 g nZVI–BC into the above solution, it was heated to 30 °C and stirred for 2 h, followed by the dropwise addition of 80 mL glutaraldehyde (2.5%, v/v). Then the mixture was stirred for 1 h at 40 °C. NaOH solution was added dropwise until the pH value reached 9 and was stirred for another 0.5 h at room temperature. Black powder was collected by filtration, washed three times with deionized water, and vacuum-dried at −25 °C for 24 h. The resulting material was referred to as C–nZVI–BC (Figure 1).

### 2.3. Characterizations

FTIR spectra were recorded on a Bruker Vector 33 spectrophotometer in the range 4000–400 cm^−1^. Samples were dried and then mixed with KBr followed by compression. Powder X-ray diffraction (PXRD) intensities were measured on a Rigaku D/max-IIIA diffractometer (Cu Kα) at 298 K. The samples were placed on a grooved aluminum plate, and patterns were recorded from 3° to 60° at a rate of 15°/min. The obtained XRD patterns were analyzed with PDXL software after removal of the background radiation. The thermal properties were measured using a gravimetric analyzer (Netzsch TG 209 F1) under a constant flow of dry nitrogen gas at a rate of 5 °C/min. An N_2_ adsorption isotherm was performed with an automatic volumetric adsorption apparatus (Autosorb-iQ, Quantachrome) at 77 K. To understand the surface characteristics and compositional distribution analysis of adsorbents, the adsorbents were studied by scanning electron microscopy combined with energy dispersive X-ray spectroscopy (SEM–EDS) (LEO1530VP, ZEISS). Samples were coated with a thin film of platinum and then imaged using a field emission operating at 20 keV. Cr element analysis was conducted after final dilution (0.1% HNO_3_: sample = 1:5) by inductively coupled plasma atomic emission spectrometry (ICP-AES) (Perkin Elmer Plasma 3200Rl). X-ray photoelectron spectroscopy (XPS) spectra were performed in a Kratos Axis ultra (DLD) spectrometer equipped with an Al Kα X-ray source in ultrahigh vacuum (UHV; <10^−10^ Torr). The surfaces of samples were cleaned by heat treatment at 100 °C in an UHV prior to the measurements.

### 2.4. Sorption Experiments

Batch experiments were conducted to evaluate the sorption capacity of the BC, nZVI–BC, and C–nZVI–BC. Additionally, the factors that influence sorption performance, such as the sorption time, the pH value, the concentration of the hexavalent chromium solution, and ionic strength, were discussed. Solutions with different Cr(VI) concentrations were prepared (50, 100, and 200 mg/L). The initial pH value of the solution was adjusted by using HNO_3_ (0.1 M) and NaOH (0.1 M) aqueous solution. The optimal sorption conditions were discussed, and the specific values of the variable are listed in Appendix A. 

In all experiments, the vials were agitated on a reciprocating shaker (160 rpm) at 30 °C for 24 h. The mixture suspensions were separated by centrifugation (4000 rpm) for 15 min. Afterwards, the adsorbents were filtered through 0.45-µm pore size filters (GE cellulose nylon membrane). The 10-mL filtrate was then treated with 2 mL 0.1% HNO_3_ solution, and Cr(VI) concentrations were analyzed. Cr(VI) concentrations were determined via inductively coupled plasma atomic emission spectrometry (ICP-AES). The Cr(VI) removal was calculated from the initial minus the final aqueous concentrations. All experiments were carried out in duplicate. 

Isotherm experiments were conducted by mixing 1 g/L adsorbent with Cr(VI) concentrations ranging from 0 to 160 mg/L at 30 °C for 24 h. Langmuir and Freundlich models were used to fit the sorption isotherm data [32,33]. The equations for these models are as follows.

Langmuir model equation:(1)$$\frac{C_{e}}{Q_{e}}=\frac{1}{K_{L}Q_{m}}+\frac{C_{e}}{Q_{m}}$$

Freundlich model equation:(2)$$\ln Q_{e}=\ln K_{f}+N\ln C_{e}$$
where *C_e_* = equilibrium concentration (mg/L); *Q_e_* = equilibrium sorption capacity (mg/g); *Q_m_* = maximum sorption capacity (mg/g); *K_L_* = Langmuir model parameter (L/mg); *K_f_* = Freundlich model parameter (mg^(1−n)^∙L^n^/g); and *n* = sorption intensity.

Kinetic experiments were studied at different time intervals (0, 10, 30, 60, 120, 240, 480, 720, 1080, and 1440 min) at 30 °C. In each study, 100 mg of adsorbates was added to 100 mL of Cr(VI) solution (100 mg/L). Three kinetic models were used to fit the sorption kinetics data [32,33]. The equations for these models are as follows.

Pseudo-first-order model equation:(3)$$\log\left(q_{e}-q_{t}\right)=\log q_{e}-\frac{k_{1}}{2.303}t$$

Pseudo-second-order model equation:(4)$$\frac{t}{q_{t}}=\frac{t}{q_{e}}+\frac{1}{k_{2}q_{e}{}^{2}}.$$

Elovich equation:(5)$$\frac{dq_{t}}{dt}=\alpha \exp\left(-\beta q_{t}\right),$$
where *q_e_* = equilibrium sorption (mg/g); *q_t_* = time *t* sorption (mg/g); *k*_1_ = rate constant of the pseudo-first-order (1/min) model; *k*_2_ = rate constant of the pseudo-second-order (g/(mg∙min)) model; *α* = initial sorption rate (mg/(g∙min)); and *β* = desorption rate (g/mg).

### 2.5. Regeneration of C–nZVI–BC Adsorbent

In addition, the reusability of the synthesized C–nZVI–BC was tested through sorption–desorption assays: 100 mg C–nZVI–BC was added into 100 mL Cr(VI) solution (100 mg/L). After agitation (160 rpm) at 30 °C for 24.0 h, the Cr(VI)-loaded C–nZVI–BC adsorbent particles were separated from the suspensions by centrifugation at 4000 rpm for 15 min. After each experiment, the adsorbent was washed thoroughly with desorption reagent (0.1 M HNO_3_) followed by treatment with deionized water to remove the adsorbed Cr ions and was eventually dried at 80 °C. The regenerated C–nZVI–BC was reused in sorption experiments with the same procedure as described above.

## 3. Results and Discussion

### 3.1. Characterization of Adsorbents

Our strategy for the synthesis of the chitosan–nZVI-modified biochar (C–nZVI–BC) is described in Figure 1. First, the thermal stability and components of the BC, nZVI–BC, and C–nZVI–BC were analyzed by using thermogravimetric analysis (TGA). The pristine BC was stable below 400 °C, which indicated the transformation from biomass to biochar. From 400 °C to 600 °C, the biochar underwent severe weight losses, which could be a continuous carbonization process (Appendix A). By comparison, the weight loss rate of nZVI–biochar was 5.35%, and C–nZVI–BC was 6.42% at ~200 °C (Appendix A), which were ascribed to the adsorbed water molecules. Moreover, the TGA curve of C–nZVI–BC revealed two weight loss regions at approximately 204~416 °C and 416~545 °C, which could be attributed to the decomposition of chitosan [34] and the transformation of residual cellulose into biochar (Appendix A), respectively. The pristine BC exhibited less residual mass (ca. 8.7%) than nZVI–BC and C–nZVI–BC when the temperature reached 800 °C (Appendix A) due to the presence of iron species in nZVI–BC and C–nZVI–BC.

In order to assess the surface areas and pore volumes of the BC, nZVI–BC, and C–nZVI–BC, we performed N_2_ adsorption–desorption experiments (Appendix A). The surface area for BC, nZVI–BC, and C–nZVI–BC was 512.89, 748.99, and 833.10 m^2^/g, respectively. The pore volume showed a very slight variation for these three biochars, ranging from 0.46 cm^3^/g to 0.61 cm^3^/g. Overall, these results indicated that the biochar surface property profoundly changed after loading nZVI and chitosan.

The X-ray diffraction pattern (XRD) was used to study the phase structure of the BC, nZVI–BC, and C–nZVI–BC. As shown in Figure 2a, the XRD pattern of pristine BC showed a wide diffraction peak at 24.3°, which might have arisen from an amorphous carbon phase structure [12,14]. The XRD pattern of the nZVI–BC showed a series of diffraction peaks in the range of 10~60° (Figure 2a). The peaks at 42.2°, 44.9°, and 49.7° indicated the presence of zerovalent iron (JCPDS, No. 06-0696). The characteristic diffraction peaks at 20.3° and 22.6° belonged to FeO(OH) [34]. The characteristic diffraction peaks at 31.8°, 37.8°, 38.5°, 41.2°, and 52.4° belonged to Fe_3_C [35]. The rest of the diffraction peaks at 39.7°, 15.5°, 35.1°, and 18.1° corresponded to FeO (JCPDS, No. 33-0664), Fe_2_O_3_ (JCPDS, No. 39-1346), and Fe_3_O_4_ (JCPDS, No. 19-0629), respectively. The results indicated that nZVI nanoparticles were easily oxidized on the biochar surface, likely because of exposure to atmosphere during the drying process and the fast redox response of iron species [36]. The preparation of C–nZVI–BC could possibly avoid the oxidation of surface nZVI particles, because chitosan is viewed as a stabilizer for reducing agents [3,24,27,29,37]. The XRD pattern for the C–nZVI–BC showed a wide diffraction peak at 27.4°: the broad and low-intensity nature of the peak indicated that the C–nZVI–BC had an amorphous structure (Figure 2a). These results indicated that nZVI particles were successfully loaded onto the surface of the *E. crassipes*-derived biochar with chitosan to form a multiple-layer composite. 

The surface functional groups of BC, nZVI–BC, and C–nZVI–BC were investigated with FTIR spectra (Figure 2b). The absorption band at 3200~3600 cm^−1^ was due to the O–H and N–H groups’ stretching vibrations. Compared to pristine BC, it was observed that the stretching vibrations of -OH in modified BCs (i.e., nZVI–BC and C–nZVI–BC) were significantly stronger, indicating that modified BCs owned more -OH groups, possibly from adsorbed water and chitosan. In addition, the new characteristic peak at about 1376 cm^−1^ for C–nZVI–BC belonged to the stretching vibration of N–O, indicating chitosan was successfully loaded onto the nZVI–BC surface [27,37]. The rest of the characteristic peaks, at about 2923, 2870, 1623, 1441, 1235, and 1088 cm^−1^, were attributed to the stretching vibrations of C–H (aliphatic), C=C (aromatic), COOH, or C–N/O groups, indicating the multifarious diversity of functional groups on the C–nZVI–BC surface. 

An X-ray photoelectron spectroscopy (XPS) analysis was utilized to analyze the surface chemical composition of BC, nZVI–BC, and C–nZVI–BC. The X-ray photoelectron spectra of pristine BC (Appendix A, line i) revealed the presence of C 1s (284.9 eV), N 1s (400.4 eV), and O 1s (532.1 eV) on the BC surface without an iron element. The spectra for the nZVI–BC and C–nZVI–BC (Appendix A, line ii and line iii) showed the peaks of C 1s, N 1s, O 1s, and Fe 2p (ranging from 702 to 730 eV). The O 1s peaks were then deconvoluted into several fine peaks for a detailed study. Two peaks were found in the O 1s spectrum of pristine BC, which corresponded to C=O and C–OH groups at 531.21 and 532.93 eV (Figure 2d) [38]. Compared to pristine BC, a remarkable change was observed in the O 1s spectra of both nZVI–BC and C–nZVI–BC. For example, the O 1s spectra of nZVI–BC showed peaks at 532.24, 532.58, and 536.99 eV, which were assigned to the C=O, C–O–R, and Fe–O bond [37,39], respectively (Figure 2e). Moreover, the peaks at 529.36, 530.47, 531.00, and 532.54 eV indicated the Fe–O–Fe, Fe–O, C=O, and C–O–R bonds on the C–nZVI–BC surface [37,38,39], respectively (Figure 2f). In addition, slight shifts of the peaks for the C=O and C–OH bonds were observed due to the modification on the biochar (Figure 2d–f). Figure 2c shows the deconvolution of Fe 2p peaks. The high-resolution XPS Fe 2p spectrum of the nZVI–BC composite exhibited two peaks with binding energies at 711.52 and 725.12 eV, which could be assigned to Fe^3+^ 2p_3/2_ and Fe^3+^ 2p_1/2_ [40], respectively. The absence of the nZVI peak suggested that nZVI on the nZVI–BC surface was transformed into iron oxide. However, with chitosan loaded on the nZVI–BC surface, the Fe 2p spectrum of the C–nZVI–BC composite exhibited two peaks with binding energies at 706.48 and 720.02 eV, which could be assigned to Fe 2p_3_ and Fe^3+^ satellite [41], respectively. The typical peak at 706.48 eV indicated the zero-valent iron phase, demonstrating that chitosan stabilizer could inhibit nZVI oxidation. In summary, these results indicated that the desired Fe elements species were observed on the modified biochar surface.

The morphology and compositional distributions of BC, nZVI–BC, and C–nZVI–BC were studied with SEM–EDS analysis. As shown in Figure 3, pristine BC had a relatively smooth surface (Figure 3a). After the nZVI modification, several relatively large nanoparticles were observed on the surface of the biochar (Figure 3b), which might have been due to magnetic interaction between iron particles [42]. The surface of C–nZVI–BC was relatively irregular and compact, but no large particles were observed with high-magnification SEM images (Figure 3c). Additionally, the elemental compositions in BC, nZVI–BC, and C–nZVI–BC were examined with SEM–EDS. For all three biochars, carbon and oxygen were uniformly distributed. An Fe signal was detected in the nZVI–BC and C–nZVI–BC (Figure 3e,f), but the Fe distribution on the C–nZVI–BC surface was relatively more uniform than that of nZVI–BC, suggesting that the introduction of chitosan could hinder the aggregation of nZVI particles. The SEM–EDS analysis showed that stabilizing using chitosan was a successful strategy to avoid nZVI aggregation.

### 3.2. Removal of Cr(VI) by Three Biochar Adsorbents

#### 3.2.1. Effect of Solution pH 

The variation of solution pH could significantly influence the speciation of Cr(VI) and the electronegativity of adsorbents. When the pH is less than 6.8, HCrO_4_^−^ becomes the major speciation that shows certain oxidation ability, but CrO_4_^2−^ dominates at pH > 6.8 [5]. In addition, the zeta-potential of C–nZVI–BC was investigated under different pH values (Figure 4a), and the isoelectric point was located at pH 3.47. Furthermore, the pH-dependent Cr(VI) sorption performances for BC, nZVI–BC, and C–nZVI–BC were examined in a pH value range of 2~8. With the increase of the initial pH values from 2 to 8, the sorption capacity of Cr(VI) significantly decreased from 82.2 to 19.8 mg/g in C–nZVI–BC, from 60.2 to 20.3 mg/g in nZVI–BC, and from 20.6 to 4.6 mg/g in BC (Figure 4b). Consequently, the sorption performance of the three biochars was optimal at lower pH values, mainly due to the protonation of functional groups on adsorbents that could enhance electrostatic attraction for Cr(VI) ions. At high pH values, the electrostatic repulsion among CrO_4_^2−^, OH^−^, and the negatively charged layer of adsorbents impeded Cr(VI) sorption. In addition, the sorption capacity and the corresponding sorption rate of C–nZVI–BC were obviously higher than those of BC and nZVI–BC (Figure 4b). This was because chitosan could provide more coordinating hetero atoms (-OH, -NH) in the C–nZVI–BC composite, which facilitated the sorption of Cr(VI) ions [43,44]. In addition, herein, the sorption properties of various biomass feedstock biochars and *Eichhornia crassipes* biochars are compared in Appendix A. Obviously, *Eichhornia crassipes* biochar had a great advantage in the sorption of Cr(VI).

#### 3.2.2. Effect of Ionic Strength and Temperature

The sorption behavior of C–nZVI–BC was investigated in Cr(VI) solutions with varying NaNO_3_ concentrations. As shown in Appendix A, the sorption capacity of Cr(VI) gradually decreased from 82.5 to 26.19 mg/g with an increase of NaNO_3_ concentration from 0.005 to 5 M at pH 2, indicating that high ionic strength impeded the Cr(VI) sorption. This was mainly because of competitive sorption of NO_3_^−^ and Cr(VI) ions on the C–nZVI–BC surface. The temperature effect on the sorption performance of C–nZVI–BC was investigated in Cr(VI) solutions, and the experiments were carried out at different temperatures (298, 308, and 318 K). As displayed in Appendix A, it was found that the maximal sorption capacity of Cr(VI) increased from 63.6 to 45.3 mg/g with an increase in temperature from 298 to 318 K, suggesting the Cr(VI) removal process was an exothermic reaction by C–nZVI–BC.

### 3.3. Sorption Isotherms and Kinetic Study

The sorption capacities of BC, nZVI–BC, and C–nZVI–BC were evaluated using Langmuir and Freundlich isotherm models. The corresponding results are presented in Figure 4c and Appendix A. The results indicated that the sorption capacity increased with a high concentration of Cr(VI) until a plateau was reached (Figure 4c). Additionally, according to the correlation coefficient (*R*^2^) values (Appendix A), the Langmuir model (0.994 for C–nZVI–BC, 0.998 for nZVI–BC, and 0.990 for BC) fit the experimental data better than the Freundlich model (0.959 for C–nZVI–BC, 0.926 for nZVI–BC, and 0.987 for BC), indicating a monolayer sorption process [45,46]. According to the results of the Langmuir model simulation, the maximum sorption capacities of Cr(VI) were 18.258 mg/g for BC, 34.029 mg/g for nZVI–BC, and 52.304 mg/g for C–nZVI–BC.

The sorption rate was another consideration. The sorption kinetics of Cr(VI) on BC, nZVI–BC, and C–nZVI–BC were investigated from 0 to 24 h. As shown, the sorption rates increased rapidly within the initial 200 min and then slowly grew until a plateau (Figure 4d). The pseudo-first-order, pseudo-second-order, and Elovich models were applied to evaluate the sorption kinetics of Cr(VI) on BC, nZVI–BC, and C–nZVI–BC [47,48], and the corresponding results are presented in Figure 4d and Appendix A. The results indicated that the pseudo-second-order model of adsorbents best fit the experimental data (Appendix A), with a correlation coefficient (*R*^2^) of 0.986 for BC, 0.956 for nZVI–BC, and 0.947 for C–nZVI–BC. The equilibration sorption capacity was 14.62 mg/g for BC, 43.78 mg/g for nZVI–BC, and 66.12 mg/g for C–nZVI–BC. Kinetics results indicated that the rate-controlling step was a chemisorption process.

### 3.4. Reusability of C–nZVI–BC

To evaluate the recyclability of C-nZVI-BC, which owned the highest sorption capacity among the three adsorbents, the regenerated C–nZVI–BC was applied into the next round of sorption for six cycles. As presented in Figure 5a, the removal efficiency of Cr(VI) gradually decreased with more recycling times, and it decreased from 82.2 mg/g in the first round to 43.1 mg/g in the sixth round, with an average loss of ~7.8 mg/g per cycle. The sorption performance of Cr(VI) through regenerated C–nZVI–BC in the second cycle is shown in Figure 5b. Overall, the C–nZVI–BC composite exhibited reasonable recyclability and held great potential for practical applications.

### 3.5. Sorption Mechanism 

SEM–EDS and X-ray photoelectron spectra were used to understand the chemical nature of nZVI–BC and C–nZVI–BC after sorption of Cr(VI). First, as presented in Figure 6, SEM images of nZVI–BC (Figure 6a) and C–nZVI–BC (Figure 6b) after sorption of Cr(VI) showed an irregular textural structure similar to that without Cr(VI) sorption (Figure 3b,c). Element mapping was used to verify the presence and distribution of Cr elements over the nZVI–BC and C–nZVI–BC surfaces. The Cr distribution was uniform and continuous over the C–nZVI–BC surface (Figure 6d); however, several Cr element aggregate peaks were observed on the nZVI–BC surface, probably due to the aggregate particles of nZVI (Figure 6c). The above results confirmed the successful sorption of Cr(VI) ions on the nZVI–BC and C–nZVI–BC surfaces.

Chemical states of the elements on the surface of nZVI–BC and C–nZVI–BC after the sorption of Cr(VI) were analyzed by XPS. Additional peaks at 577.4 eV (Cr 2p_3/2_) and 587.4 eV (Cr 2p_1/2_) were observed [27,29,33] (Appendix A) when spectra were compared to nZVI–BC and C–nZVI–BC without Cr(VI) sorption (Appendix A) [49,50], which corresponded to adsorbed Cr ions. The broad Cr 2p_3/2_ peak could be divided into two peaks at 579.18 eV (Cr(VI)) and 577.32 eV (Cr(III)) for nZVI–BC (Figure 7c) and 578.84 eV (Cr(VI)) and 576.85 eV (Cr(III)) for C–nZVI–BC (Figure 7d). Besides, deconvolution of the Cr 2p_1/2_ peak resulted in two components, included 590.7 eV (Cr(VI)) and 587.21 eV (Cr(III)) for nZVI–BC (Figure 7c) and 589.45 eV (Cr(VI)) and 587.01 eV (Cr(III)) for nZVI–BC (Figure 7d) (Appendix A). Apparently, nZVI–BC and C–nZVI–BC not only provided sorption sites for Cr(VI), but also reduced Cr(VI) into Cr(III), mostly because of the role of nZVI. To further understand the difference between nZVI–BC and C–nZVI–BC in the sorption of Cr(VI) ions, the O 1s and Fe 2p spectra of Cr(VI) adsorbed on nZVI–BC and C–nZVI–BC were analyzed. The high-resolution XPS O 1s spectra of nZVI–BC and C–nZVI–BC could be deconvoluted into three different oxygen-containing functional groups, respectively: (i) O−M (H, Fe, and Cr) at 530.52 eV, (ii) C=O at 531.65 eV, and (iii) C–O–R at 532.83 eV for nZVI–BC (Figure 7a); and (i) M–O–M at 529.88 eV, (ii) C=O at 531.15 eV, and (iii) C–O–R at 532.96 eV for C–nZVI–BC (Figure 7b) [50,51]. The sorption mechanism of Cr(VI) on C–nZVI–BC was further studied because of the preeminent sorption performance. It was found that the peak intensity of metal O species was significantly increased in comparison to pure nZVI–BC and C–nZVI–BC (Figure 2e,f and Figure 7a,b), suggesting the formation of a bond between oxygen-containing groups and the Cr ion. Besides, shifts of the Fe 2p peaks from 706 eV to 711 eV were observed when they were compared to the spectrum of the pure C–nZVI–BC. Deconvolution of the Fe 2p peaks gave excellent fittings and showed six components, that is, Fe_2_O_3_ at 711.85 and 724.60 eV, FeO at 715.13 and 727.65 eV, and Fe^3+^ satellites at 719.75 and 732.82 eV (Figure 7f) [40,41]. Collectively, the XPS results suggested that the sorption process of Cr(VI) onto C–nZVI–BC underwent the following steps: the coordination of O with Cr to form Cr–O complexes and the reduction of Cr(VI) into Cr(III). Considering all of the evidence presented above, we herein propose four mechanisms for Cr(VI) sorption on the C–nZVI–BC surface (Figure 8), that is, complexation, precipitation, electrostatic interactions, and chemical reduction.

## 4. Conclusions

In summary, a low-cost and eco-friendly adsorbent with high sorption capacity was fabricated from *E. crassipes* biomass. Among BC, nZVI–BC, and C–nZVI–BC, C–nZVI–BC showed the best sorption performance. The maximum sorption capacity of Cr(VI) by C–nZVI–BC was 82.8 mg/g at pH 2, much higher than pristine BC (20.6 mg/g) and nZVI–BC (60.2 mg/g). Additionally, the sorption capacity of Cr(VI) was increased at a lower pH and ionic strength. Batch Cr(VI) sorption experiments were well fitted by the Langmuir and pseudo-second-order kinetic models, suggesting a monolayer chemical sorption process for all three adsorbents. A reusability assay indicated that C–nZVI–BC could be reused for six cycles, and more than 43% sorption capacity was retained. The surface chemical nature of the Cr(VI)-adsorbed C–nZVI–BC was investigated with XPS, suggesting four sorption mechanisms, i.e., complexation, precipitation, electrostatic interactions, and chemical reduction. This study, as a prototype demonstration, converted a biomass hazard in the environment into a highly efficient adsorbent, paving the road for future research on utilizing waste by considering environmental applications.

## Figures and Tables

**Figure 1 ijerph-16-03046-f001:**
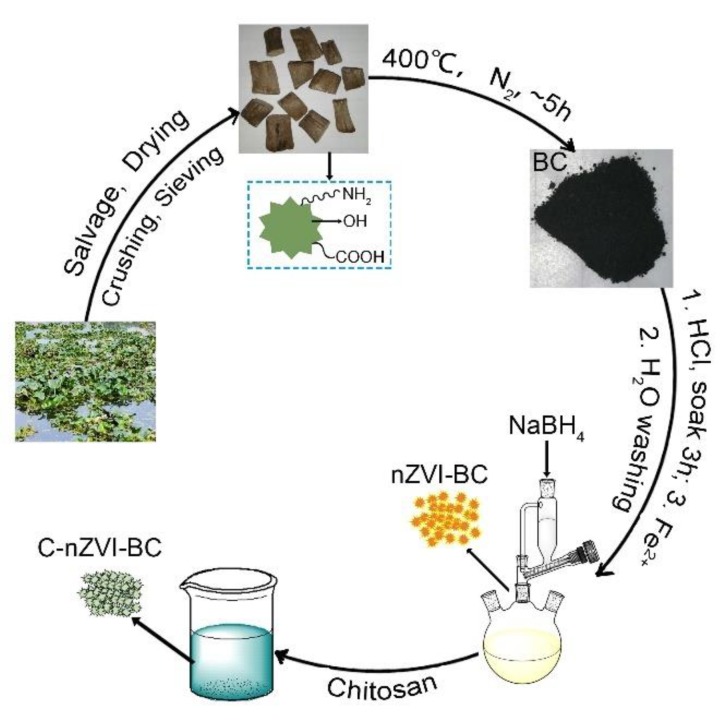
A scheme showing the process of the production of biochar (BC), nanoscale zero-valent ion (nZVI)–BC, and chitosan (C)–nZVI–BC.

**Figure 2 ijerph-16-03046-f002:**
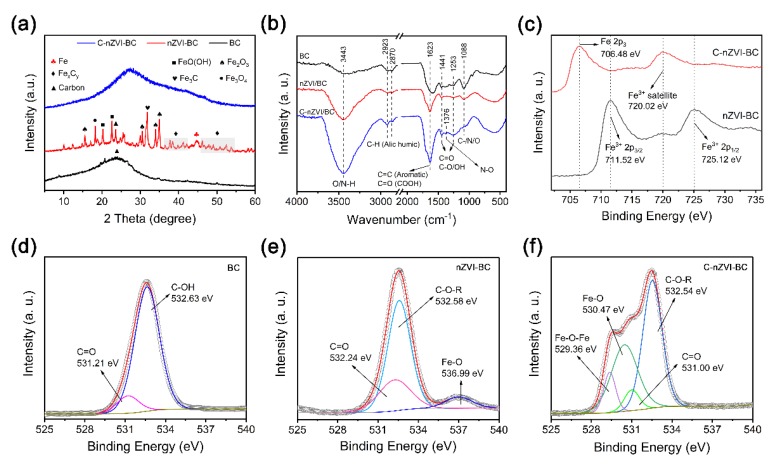
Characterization of BC, nZVI–BC, and C–nZVI–BC. Powder X-ray diffraction (PXRD) patterns (**a**); FTIR spectra (**b**); high-resolution X-ray photoelectron spectroscopy (XPS) spectra of Fe 2p for nZVI–BC and C–nZVI–BC (**c**); high-resolution XPS spectra of O 1s for BC, nZVI–BC, and C–nZVI–BC (**d**–**f**).

**Figure 3 ijerph-16-03046-f003:**
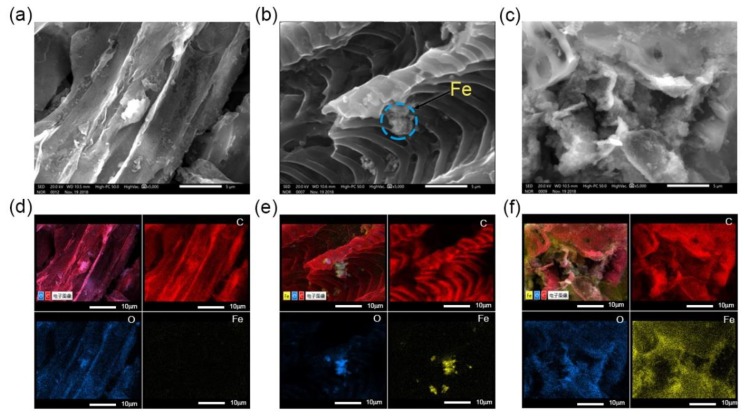
SEM microscopic images of as-prepared BC (**a**), nZVI–BC (**b**), and C–nZVI–BC (**c**). Scale bar = 5 µm. The element mapping indicated a distribution information of Fe (yellow), O (blue), and C (red) of the corresponding BC (**d**), nZVI–BC (**e**), and C–nZVI–BC (**f**). Scale bar: 10 µm.

**Figure 4 ijerph-16-03046-f004:**
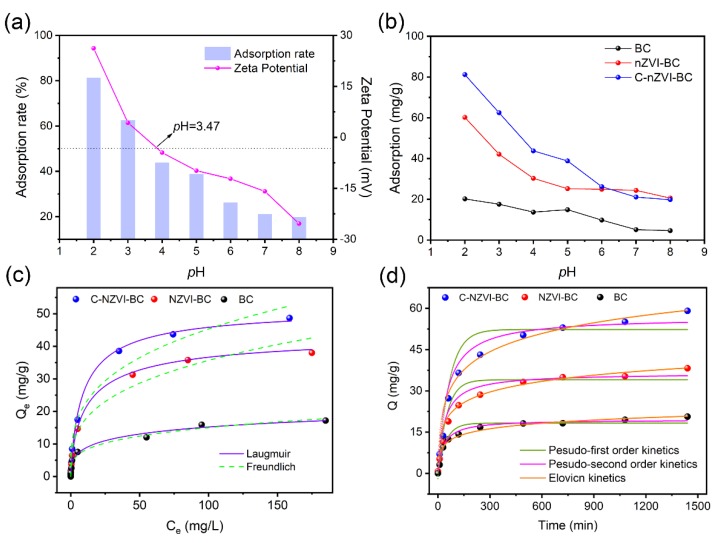
Sorption performance of hexavalent chromium (Cr(VI)) onto three adsorbents. Sorption rate and zeta potential of C–nZVI–BC in different pH solutions (**a**); effect of pH on the sorption of Cr(VI) onto adsorbents (*T* = 30 °C, *V* = 100 mL, weight of adsorbents = 100 mg, and initial Cr(VI) concentration = 100 mg/L) (**b**); Cr(VI) sorption isotherms and fitting results (**c**); and Cr(VI) sorption kinetics and fitting results (**d**) (pH = 2, shaking for 24 h).

**Figure 5 ijerph-16-03046-f005:**
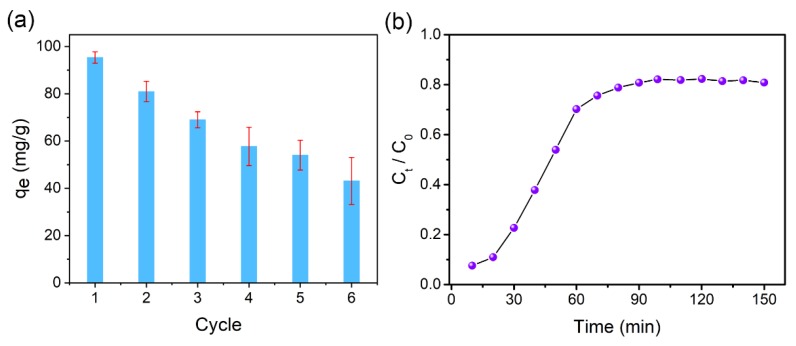
Six consecutive sorption–desorption cycles of Cr(VI) on C–nZVI–BC (**a**); sorption performance of Cr(VI) through regenerated C–nZVI–BC in the second cycle (**b**) (*C*_0_ = 100 mg/L, *V* = 100 mL, *t* = 6 h, *T* = 30 °C).

**Figure 6 ijerph-16-03046-f006:**
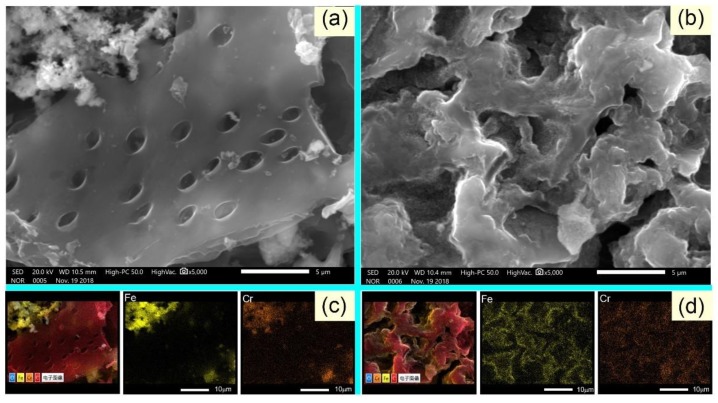
The SEM microscopic images of Cr(VI) adsorbed on nZVI–BC (**a**) and C–nZVI–BC (**b**). Scale bar = 5 µm. The energy-dispersive X-ray spectroscopy (EDS) element mapping of Fe (yellow) and Cr (golden) of the corresponding Cr(VI) adsorbed on nZVI–BC (**c**) and C–nZVI–BC (**d**). Scale bar: 10 µm.

**Figure 7 ijerph-16-03046-f007:**
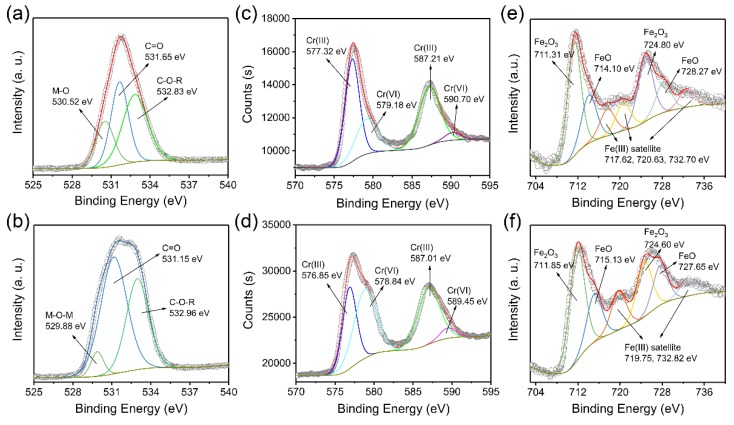
High-resolution XPS spectra of O 1s, Cr 2p, and Fe 2p for Cr(VI) adsorbed on nZVI–BC (**a**,**c**,**e**) and C–nZVI–BC (**b**,**d**,**f**).

**Figure 8 ijerph-16-03046-f008:**
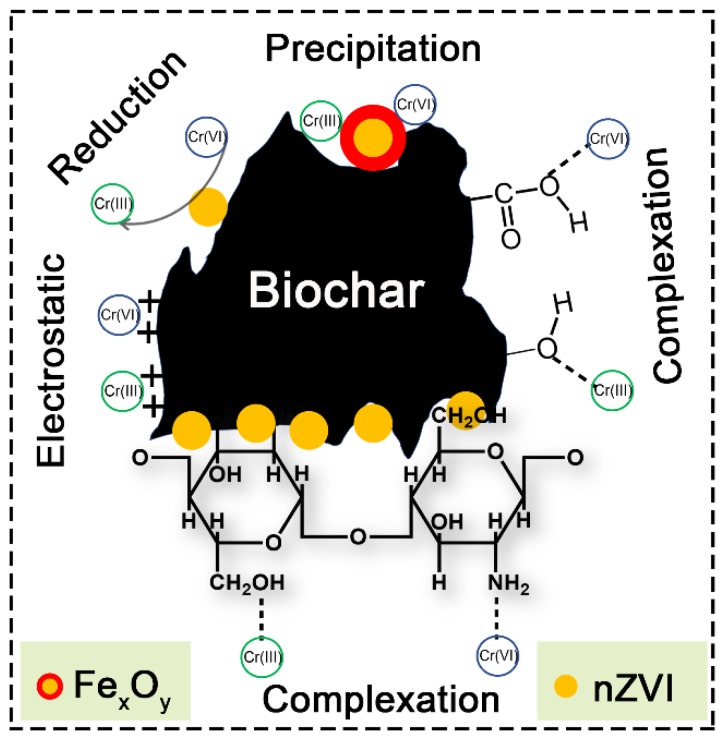
Mechanisms of Cr(VI) sorption by C–nZVI–BC.

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
