# Peer review of "Nanoscale Zero-Valent Iron and Chitosan Functionalized Eichhornia crassipes Biochar for Efficient Hexavalent Chromium Removal"

_ijerph, 2019, doi:10.3390/ijerph16173046_

Round 1
Reviewer 1 Report
This study may be interesting to some researchers but there are several deficiencies that need to be resolved before it can be accepted. The authors used XPS to speciate sorbed Cr on the surface of nZVI-BC and C-nZVI_BC but did not present semi-quantitative information, i.e. how much (in percentage) of the sorbed Cr is in the Cr(VI) form and how much in the Cr(III) form. A table can be added to list the results. The XRD data presented in Figure 2a need to be explained in more detail. Why all the XRD peaks that were present in nZVI-BC for elemental iron, iron oxides, and Fe3C disappeared in the C-nZVI-BC sample? Did chitosan and/or glutaraldehyde act as a chemical reductant? Why glutaraldehyde was used in the synthesis? The word "adsorption" through the text needs to be changed to "sorption" or "immobilization" that includes adsorption, surface complexation, and surface precipitation. Page 1, L15, change "absorbents" to "sorbents".Author Response
This study may be interesting to some researchers but there are several deficiencies that need to be resolved before it can be accepted.
Q1.The authors used XPS to speciate sorbed Cr on the surface of nZVI-BC and C-nZVI_BC but did not present semi-quantitative information, i.e. how much (in percentage) of the sorbed Cr is in the Cr(VI) form and how much in the Cr(III) form. A table can be added to list the results.
Answer: Thanks for the question. According to Deconvolution of Cr 2p peaks data, we obtained that the semi-quantitative informationof Cr(III) and Cr(VI)( See Table S6 SI)
|
|
Cr(III) |
Cr(VI) |
|
nZVI-BC |
74.74 |
25.26 |
|
C-nZVI-BC |
59.95 |
40.05 |
Q2.The XRD data presented in Figure 2a need to be explained in more detail. Why all the XRD peaks that were present in nZVI-BC for elemental iron, iron oxides, and Fe3C disappeared in the C-nZVI-BC sample?
Answer: Thanks for the question and careful check. Because 1) chitosan could avoid the oxidation of surface nZVI particles, so iron oxides disappeared in the C-NZVI-BC sample; 2) the element iron species peak was embedded within amorphous carbon peak. Please check line 197-200.
Q3.Did chitosan and/or glutaraldehyde act as a chemical reductant?
Answer: Thanks for the question. Chitosan acts as a stabilizing agent. And glutaraldehyde acts as a chemical reductant and fixing agent.
Q4.Why glutaraldehyde was used in the synthesis?
Answer: Thanks for the question. Glutaraldehyde acts as a chemical reductant and fixing agent.
Q5.The word "adsorption" through the text needs to be changed to "sorption" or "immobilization" that includes adsorption, surface complexation, and surface precipitation.
Answer: Thanks for the careful check. The word “adsorption” of the manuscript has been changed.
Q6.Page 1, L15, change "absorbents" to "sorbents".
Answer: Thanks for the careful check. The word “adsorbents” of the manuscript has been changed.
Reviewer 2 Report
General Comments
Manuscript No: ijerph-558412 presents the synthesis of nZVI on biochar and evaluation its capacity for chromium removal. This experimental work is well described and presented with details.
There is not sufficiently detailed discussion and comparison with previously published data on the synthesis of iron nanocomposites for chromium removal. Authors should mention in a paragraph before conclusion other studies such as: Toli A., Varouxaki A., Mystrioti C., Xenidis A., Papassiopi Ν. 2018, Green Synthesis of Resin Supported Nanoiron and Evaluation of Efficiency for the Remediation of Cr(VI) Contaminated Groundwater by Batch Tests. Bulletin of Environmental Contamination and Toxicology, 101(6):711-717. DOI: 10.1007/s00128-018-2425-2C and compare them with their results.
Author Response
General Comments
Manuscript No: ijerph-558412 presents the synthesis of nZVI on biochar and evaluation its capacity for chromium removal. This experimental work is well described and presented with details.
Answer: Thank you for your positive comment on our work, please check our response below.
There is not sufficiently detailed discussion and comparison with previously published data on the synthesis of iron nanocomposites for chromium removal. Authors should mention in a paragraph before conclusion other studies such as: Toli A., Varouxaki A., Mystrioti C., Xenidis A., Papassiopi Ν. 2018, Green Synthesis of Resin Supported Nanoiron and Evaluation of Efficiency for the Remediation of Cr(VI) Contaminated Groundwater by Batch Tests. Bulletin of Environmental Contamination and Toxicology, 101(6):711-717. DOI: 10.1007/s00128-018-2425-2C and compare them with their results.
Answer: Thanks for the suggestion. We have added the relative reference comparison results in the manuscript.
Reviewer 3 Report
1- Line 80: write calcined and not calcinated and the whole text needs to be checked
2- It would be interesting to study the effect of temperature on the adsorption process
3- Compare the findings of this study with those obtained in literature for other adsorbents derived from biomass
Author Response
1- Line 80: writecalcinedand not calcinated and the whole text needs to be checked
Answer: Thanks for the careful check. We have modified “calcined” in the whole text.
2- It would be interesting to study the effect of temperature on the adsorption process
Answer: Thanks for the suggestion. We have checked and updated the data of temperature effect on the adsorption of Cr(VI) onto C-NZVI-BC, and the data have been added in the Supporting information (See Figure S6 SI).
3- Compare the findings of this study with those obtained in literature for other adsorbents derived from biomass
Answer: Thanks for the suggestion. A table can be added to list the different biomass feedstock biochars and Eichhornia crassipesbiochar (See Table S7).
Reviewer 4 Report
This manuscript describes a method for the preparation of low cost and environmental-friendly chromium ion adsorbent by nano-size iron particles loaded and chitosan stabilized biochar. The results are interesting and the work can be published in this journal.
Line 87, check the “0.1 mmol, 287 mg”. The molecular weight is 287, and 1.0 mmol of the compound should be 287 mg in weight.
Author Response
This manuscript describes a method for the preparation of low cost and environmental-friendly chromium ion adsorbent by nano-size iron particles loaded and chitosan stabilized biochar. The results are interesting and the work can be published in this journal.
Answer: Thanks for your positive comment, please check our response below.
Line 87, check the “0.1 mmol, 287 mg”. The molecular weight is 287, and 1.0 mmol of the compound should be 287 mg in weight.
Answer: Thanks for the review and careful check. We have corrected it.
Round 2
Reviewer 1 Report
The revisions seem to be satifactory.